# CHANNEL EQUILIBRIUM NETWORKS

## ABSTRACT

Convolutional Neural Networks (CNNs) typically treat normalization methods such as batch normalization (BN) and rectified linear function-like activations (*e.g.* ReLU) as building blocks. Previous work pointed out that learning feature channels with equal magnitudes is important for a CNN to achieve good generalization ability. However, the above "Norm+ReLU-like" basic block often learns inhibited channels that has small magnitude (*i.e.* contributes little to the feature representation), impeding both learning and generalization ability of CNNs. This problem is seldom explored in the literature. To mitigate the inhibited channels and encourage channels to contribute equally to the feature representation, we propose a new building block, Channel Equilibrium (CE), which is able to prevent inhibited channels in both experiments and theory. CE has several appealing properties. First, CE can be stacked after many different normalization methods such as BN and Group Normalization (GN), as well as integrated into many advanced CNN architectures such as ResNet and MobileNet V2 to form a series of CE networks (CENets), outperforming existing network architectures. Second, CE has an interesting connection with the Nash Equilibrium, a well-known solution of a non-cooperative game. Third, extensive experiments show that CE achieves state-of-the-art results on various challenging benchmarks such as ImageNet and COCO. The models and codes will be released.

## 1 INTRODUCTION

Normalization is an important technique for a wide range of tasks such as image classification (Ioffe & Szegedy, 2015), object detection (He et al., 2017a; Wu & He, 2018), and image generation (Miyato et al., 2018). In recent years, a lot of work improved normalization methods, such as batch normalization (BN) (Ioffe & Szegedy, 2015), layer normalization (LN) (Ba et al., 2016) and switchable normalization (SN) (Luo et al., 2018). These methods are often used together with the ReLU-like activation functions such as ReLU (Glorot et al., 2011; Nair & Hinton, 2010), ELU (Clevert et al., 2015) and Leaky ReLU (LReLU) (Maas et al., 2013), making the "Norm+ReLU-like" module become one of the most widely-used building blocks of modern CNNs. This work investigates and alleviates the inhibited channels emerged in the "Norm+ReLU-like" building block, which consists of a normalization layer and a ReLU-like activation function given by

$$y_{ncij} = g(\tilde{x}_{ncij}), \qquad \tilde{x}_{ncij} = \gamma_c \bar{x}_{ncij} + \beta_c, \tag{1}$$

where subscript $n, c, i$, and $j$ denote indices of a sample, a channel, height and width of a feature channel respectively. For instance, $y_{ncij}$ indicates output value of the location $(i, j)$ in the $c$-th channel of the $n$-th sample. And $\tilde{x}_{ncij}$ and $\bar{x}_{ncij}$ represent normalized channel features and standardized channel features respectively. $\gamma$ and $\beta$ are two vectors of parameters, where each element re-scales and re-shifts the standardized features for each channel $c$. Moreover, $g(\cdot)$ denotes a ReLU-like activation function.

As pointed out in (Morcos et al., 2018), a CNN that generalizes well would have its channels contributed equally in its feed-forward computations. We term this desired property "channel equalization". However, a recent study disclosed a critical problem of the "Norm+ReLU-like" builiding block, known as "channel collapse", where certain channels always produce small output values given any input. For example, the lottery hypothesis (Frankle & Carbin, 2018) claimed that one over-parameterized CNN always contains unimportant channels that contribute little to the network's prediction. This paper shows that these unimportant channels are *inhibited channels*, which usually

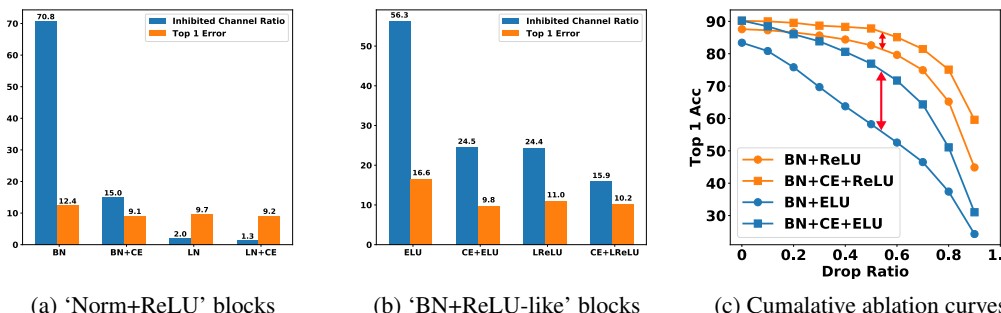

(a) 'Norm+ReLU' blocks     (b) 'BN+ReLU-like' blocks     (c) Cumalative ablation curves

Figure 1: We train VGGNet on CIFAR-10 with various 'Norm+ReLU' blocks (details about training settings are provided in Sec.F of Appendix). The inhibited channel ratio is defined as the average percentage of values less than 1e-2 in the first six feature maps of VGGNet. (a) & (b) show that inhibited channels emerge in many 'Norm+ReLU-like' blocks such as 'BN+ReLU', 'LN+ReLU' and 'BN+ELU'. (c) plots cumulative ablation curves that is a technique demonstrating channel equalization (Morcos et al., 2018). Equipped with CE, both 'BN+ReLU' and 'BN+ELU' presents a more gentle drop curve of top-1 accuracy versus cumulative ablation of channels, implying that CE helps channel equalization.

associated with small values of $\gamma$ in Eqn.(1) or small values of the channel features (In this paper, inhibited channel is defined as channel whose all feature values are less than $1e - 2$). For example, as shown in Fig.1(a,b), the inhibited channels exist in many "Norm+ReLU-like" basic block such as 'LN+ReLU', 'BN+ELU' and 'BN+ReLU'. This phenomenon has motivated many investigations to directly remove these inhibited channels, such as network slimming (Liu et al., 2017; Yu et al., 2018) and channel pruning (He et al., 2017b). However, disequilibrium among channels caused by the inhibited channels would do harm to generalization ability of the network and simply removing the inhibited channels that are inactive during training does not help improve learning capacity.

Instead of removing the inhibited channels, we present an alternative perspective by proposing a novel building block, named Channel Equilibrium (CE), to recover and equalize the inhibited channels by encouraging channels to contribute equally in the feature representation learning process, thus enhancing the representation and generalization of CNNs. To this end, a key observation from Eqn.(1) is that the dependency (covariance) matrix of all the feature channels after normalization is scaled by $\gamma\gamma^\mathsf{T}$. Let this covariance matrix be $\Sigma$. We will see that by applying a decorrelation operator, *i.e.* $\Sigma^{-\frac{1}{2}}$, we can not only effectively eliminate the correlations between computations of channel features and the magnitude of $\gamma$, but also equalize the channels' magnitudes (Barlow et al., 1961; Bengio & Bergstra, 2009). This operator enables all the channels to play an equal role in the computations of a CNN, improving its generalization ability. For example, as shown in Fig.1, the VGGNet (Simonyan & Zisserman, 2014) equipped with CE is able to effectively prevent inhibited channels and achieves channel equalization in various of "Norm+RelU-like" blocks, consistently improving their recognition performance.

The main **contributions** of this work are three-fold. First, we introduce an efficient and effective building block, Channel Equilibrium (CE), which encourages channel equalization and enhances representation learning of CNNs. Second, CE blocks can be stacked after common normalizers and plugged into various advanced architectures, consistently improving their performance by a large margin. For example, CE can be integrated into ResNet and MobileNet V2, forming a series of CE-Networks by replacing the ordinary 'BN-ReLU' block by 'BN-CE-ReLU' block, which merely introduces subtle extra computational complexity. As a result, the CE-ResNet50 and CE-MobileNet V2 outperform their counterparts by 1.7% and 2.1% top-1 accuracy with nearly the same FLOPs. We also show that combining CE with synchronization across GPUs increases the AP metric on the MS-COCO dataset to 42.0, surpassing its counterpart by 3.4. Third, the learned equalized feature representation of CENet can be better transferred to many other tasks like object detection and segmentation.

## 2 RELATED WORK

**Channel equalization.** Channel equalization indicates that channels in a layer of CNNs contribute equally to network's computation. The success of the two most commonly used regularization techniques, i.e. BN (Ioffe & Szegedy, 2015) and Dropout (Srivastava et al., 2014), is attributed to channel or neuron equalization. For example, Mianjy et al. (2018) showed that Dropout makes the norm of incoming/outgoing weight vectors of all the hidden nodes equal, indicating a kind of equalization between neurons. Moreover, Morcos et al. (2018) pointed out that BN implicitly discourages single direction reliance, indicating that equalizing different channels is able to enhance the generalization of learned feature representation. Note that the squeeze-and-excitation (SE) network (Hu et al., 2018) is a pioneer work that explicitly modeling interdependencies among channels by investigating network design. However, SE selectively emphasizes informative channels and suppresses less useful ones. In contrast, the proposed CE block encourages all the channels to play an equal role in network's computation, which, as will be shown, can be linked with Nash Equilibrium. More related work on sparsity in ReLU and normalization methods are provided in Sec.A of Appendix.

## 3 METHOD

In this section, we first review the normalization method and then introduce the proposed Channel Equilibrium (CE) block. CE contains two complementary branches, i.e. batch decorrelation (BD) and adaptive instance inverse (AII). We show how BD and AII benefit from each other through parameter $\gamma$ and how CE is linked with Nash Equilibrium.

**Notations.** For CNNs, we use $x \in \mathbb{R}^{N \times C \times H \times W}$ to represent the feature in a layer, specifically, $x_{ncij}$ denotes a pixel $(i, j)$ in the $c$-th channel of the $n$-th sample. Sometimes, we ignore the subscript '$n$' and denote it as $x_{cij}$ for clarity of notation. $x_{nij} \in \mathbb{R}^C$ is obtained by stacking elements in all channels of $x_{ncij}$ into a column vector. $\text{Diag}(\cdot)$ returns a matrix with the given diagonal and zero off-diagonal entries, and $\text{diag}(\cdot)$ extracts the diagonal of the given matrix. $\gamma, \beta \in \mathbb{R}^C$ are normalization parameters.

### 3.1 OVERVIEW OF NORMALIZATION

Normalization is usually employed after convolution layers to stabilize the training of CNNs. Given a hidden feature $x \in \mathbb{R}^{N \times C \times H \times W}$, a normalizer first standardizes it to $\bar{x}$, and then maps it to $\tilde{x}$ by an affine transformation, as written by

$$\tilde{x}_{ncij} = \gamma_c \bar{x}_{ncij} + \beta_c, \qquad \bar{x}_{ncij} = (x_{ncij} - \mu_s)/\sigma_s \qquad (2)$$

where $s \in \Omega = \{\text{IN}, \text{BN}, \text{LN}, \cdots\}$ indicates a normalizer and $\mu_s$, $\sigma_s$ are the mean and standard deviation of the given normalizer. For simplicity, $\epsilon$ in original formulation is omitted (Ioffe & Szegedy, 2015). From Eqn.(2), we claim that normalization would lead to an unequal feature representation in a channel basis, based on the fact that common-used normalizers like IN and BN are performed channel-wisely. It is known that importance of channels are quantified by the learned parameters $\gamma$ with its magnitude since channel features are scaled by $\gamma$ in a channel basis. Previous work (Frankle & Carbin, 2018; Mehta et al., 2019) revealed that some inhibited channels emerge when the associated $\gamma$ or feature map gets small (Mehta et al., 2019; Lu et al., 2019). Obviously, the inhibited channels would cause disequilibrium among channels, resulting in limited generalization ability. To alleviate such a disequilibrium, Channel Equilibrium (CE) block is proposed in the following section.

### 3.2 CHANNEL EQUILIBRIUM (CE) BLOCK

A Channel Equilibrium (CE) block is a computational unit which aims to equalize feature representation capacity among channels. To this end, decorrelation method is adopted. Different from previous methods (Huang et al., 2018; 2019) decorrelating features by a single batch estimated covariance matrix $\Sigma$, the proposed method brings in an adaptive instance variance, $S_n$, on the diagonal of the covariance matrix $\Sigma$, considering that channel dependency is specific to each input (Hu et al., 2018), as formulated in the following,

$$D_n = \lambda \Sigma + (1 - \lambda)\text{Diag}(S_n), \qquad S_n = F(\sigma^2(\tilde{x}_n)), \qquad (3)$$

where the subscript $n$ is the sample index, $\lambda \in (0, 1)$ is a trainable ratio used to switch between batch and instance statistics, $F : \mathbb{R}^C \to \mathbb{R}^C$ is a transformation conditioned on the current input $\tilde{x}$ and $\sigma^2(\tilde{x}_n)$ computes instance variance of $\tilde{x}_n$ within each channel. On the issue of channel disequilibrium, CE block works by decorrelating feature maps after normalization using $D_n^{-\frac{1}{2}}$. Further, the Jensen inequality for matrix functions (Pečarić, 1996) can be employed to obtain a relaxed decorrelation operator $D_n^{-\frac{1}{2}}$:

$$D_n^{-\frac{1}{2}} = [\lambda \Sigma + (1-\lambda)\mathrm{Diag}(S_n)]^{-\frac{1}{2}} \preceq \lambda \Sigma^{-\frac{1}{2}} + (1-\lambda)\left[\mathrm{Diag}(S_n)\right]^{-\frac{1}{2}}, \tag{4}$$

where $A \preceq B$ indicates $B - A$ is semi-definite positive. We introduce this relaxation for the following two reasons. (1) Computation reduction. It allows less computational cost for each training step since the relaxed form only needs to calculate the inverse of square root $\Sigma^{-\frac{1}{2}}$ once, and the other branch $\mathrm{Diag}(S_n)^{-\frac{1}{2}}$ is easy to compute. (2) Inference acceleration. $\Sigma^{-\frac{1}{2}}$ is a moving-average statistic in inference which can be absorbed into previous layer, therefore, enabling fast inference. Note that Eqn.(4) transforms the combination of covariance and adaptive instance variance into the combination of their inverse square roots.

In the following, we refer $\Sigma^{-\frac{1}{2}}$ in Eqn.(4) as batch decorrelation (BD) and refer $\left[\mathrm{Diag}(S_n)\right]^{-\frac{1}{2}}$ as adaptive instance inverse (AII). The former decorrelates channels by a batch covariance, while the latter adjusts the extend of inverse for each channel and instance in an adaptive manner. Integrating both of them yields the forward representation of CE block:

$$p_{nij} = D_n^{-\frac{1}{2}}(\mathrm{Diag}(\gamma)\bar{x}_{nij} + \beta) \tag{5}$$

where $p_{nij} \in \mathbb{R}^C$ denotes the output of CE, as illustrated in Fig.2(b). Note that CE is performed after the normalization layer, BN is taken as an example to introduce these two branches in the following sections.

### 3.2.1 Batch Decorrelation (BD)

Although a lot of previous work (Huang et al., 2018; 2019; Pan et al., 2019) has investigated whitening method using covariance matrix, all of them are applied after the convolution layer. Thus, inhibited channels still exist since whitened channel features are also scaled by $\gamma$. Instead, decorrelation in CE is applied after the normalization layer to equalize magnitude of all channels. Consider a tensor $\tilde{x}$ after a BN layer, it can be reshaped as $\tilde{x} \in \mathbb{R}^{C \times M}$ and $M = N \cdot H \cdot W$. Then the covariance matrix $\Sigma$ of $\tilde{x}$ can be written as (details are presented in Sec. B of Appendix)

$$\Sigma = \gamma\gamma^\mathsf{T} \odot \frac{1}{M}\bar{x}\bar{x}^\mathsf{T} \tag{6}$$

where $\bar{x}$ is a standardized feature with zero mean and unit variance and $\odot$ indicates elementwise multiplication. Eqn.(6) implies that the covariance matrix $\Sigma$ of $\tilde{x}$ can be decomposed into two parts. The first part depends on normalization parameter $\gamma$ and the second part becomes correlation matrix of $\tilde{x}$. It is observed that $\Sigma_{ij}$, which represents dependency between $i$-th channel and $j$-th channel, is scaled by $\gamma_i\gamma_j$ after BN is applied.

The Batch Decorrelation (BD) branch requires computing $\Sigma^{-\frac{1}{2}}$, which is usually related to eigen-decomposition or SVD and involves heavy computation (Huang et al., 2018). Instead, here we adopt an efficient approach, i.e., Newton's Iteration to obtain $\Sigma^{-\frac{1}{2}}$ (Bini et al., 2005; Higham, 1986). Given covariance matrix $\Sigma$, Newton's Iteration calculates $\Sigma^{-\frac{1}{2}}$ by the following iterations:

$$\begin{cases} \Sigma_0 = I \\ \Sigma_k = \frac{1}{2}(3\Sigma_{k-1} - \Sigma_{k-1}^3\Sigma), \ k = 1, 2, \cdots, T. \end{cases} \tag{7}$$

where $T$ is the iteration number ($T = 3$ in our experiments). Note that the convergence of Eqn.(7) is guaranteed if $\|\Sigma\|_2 < 1$ (Bini et al., 2005). To this end, $\Sigma$ is normalized as $\Sigma/\mathrm{tr}(\Sigma)$ where $\mathrm{tr}(\cdot)$ is the trace operator (Huang et al., 2019). In this way, the normalized covariance matrix is written as $\Sigma = \frac{\gamma\gamma^\mathsf{T}}{\|\gamma\|_2^2} \odot \frac{1}{M}\bar{x}\bar{x}^\mathsf{T}$. To sum up, the batch decorrelation branch firstly calculates a normalized covariance matrix and then applies Newton's Iteration to obtain its inverse square root, reducing lots of computational cost compared with SVD decomposition in the training stage. Furthermore, BD branch can be merged into convolutional layers in the inference stage, which adds extra computation marginally.

### 3.2.2 ADAPTIVE INSTANCE INVERSE (AII)

Channel dependencies are specific to each sample. Consequently, a conditional decorrelation is desired for each sample. The adaptive instance inverse (AII) branch only uses diagonal entries to model channel dependencies, as shown in Eqn.(3). Since a diagonal matrix can be inverted easily, this approach can avoid the computation of Eqn.(4).

To construct the AII branch, we analyze its input (the output of a BN layer), which is formulated as $\tilde{x}_{ncij} = \gamma_c \bar{x}_{ncij} + \beta_c$. The input of AII is the instance variance of each channel (details are provided in Appendix Sec. B),

$$\sigma_{nc}^2 = \frac{\gamma_c^2 (\sigma_{\mathrm{IN}}^2)^{nc}}{(\sigma_{\mathrm{BN}}^2)^c} \tag{8}$$

where $\sigma_{\mathrm{IN}}^2$ and $\sigma_{\mathrm{BN}}^2$ represent the variances in IN and BN respectively. The ratio of them measures the relative fluctuation of how much the instance statistic are deviated from the batch-estimated statistic. Similar to Eqn.(6), the input of AII is also scaled by $\gamma_c^2$.

The AII branch takes $\sigma_{nc}^2$ as input and computes an adaptive instance inverse, i.e. $[\mathrm{Diag}(S_n)]^{-\frac{1}{2}}$. It needs to satisfy two requirements. First, as is desired, dependencies among channels should be embedded in $[\mathrm{Diag}(S_n)]^{-\frac{1}{2}}$ for each sample. Second, the output of AII should have the same philosophy as inverse square root of variance or covariance in BD branch. To achieve this, a reparameterization trick is employed to generate adaptive instance inverse. Let $s$ be an estimate of variance, the AII branch can be reparameterized as below,

$$[\mathrm{Diag}(S_n)]^{-\frac{1}{2}} = \mathrm{Diag}(\tilde{F}(\sigma_n^2)) \cdot s^{-\frac{1}{2}}, \tag{9}$$

$$\tilde{F}(\sigma_n^2) = \delta_2(W_2 \delta_1(\mathrm{LN}(W_1 \sigma_n^2))), \quad s = \sigma^2(\tilde{x}), \tag{10}$$

where $\delta_1$ and $\delta_2$ are ReLU and sigmoid activation function respectively, $W_1 \in \mathbb{R}^{\frac{C}{r} \times C}$ and $W_2 \in \mathbb{R}^{C \times \frac{C}{r}}$ and $r$ is reduction ratio, $s \in \mathbb{R}$ denotes variance of all elements in $\tilde{x}$, which is a batch statistic in training and is obtained using moving average for inference. $\tilde{F}(\sigma_n^2) \in (0, 1)^C$ is treated as a gating mechanism in order to control the strength of instance inverse for each channel. Following the best practice (Hu et al. (2018); Cao et al. (2019)) in characterizing channel relationships, $\tilde{F}$ is expressed by a bottleneck architecture that is able to model channel dependencies and limit model complexity. Layer normalization (LN) is used inside the bottleneck transform (before ReLU) to ease optimization. It is seen from Eqn.(9) that $s^{-\frac{1}{2}}$ represents the quantity of inverse square root of variance and $\tilde{F}(\sigma_n^2)$ regulates the extend of variance inverse. $\tilde{F}$ maps the instance variance to a set of channel weights. In this sense, the AII branch intrinsically introduces channel dependencies conditioned on each input.

### 3.3 DISCUSSIONS

**Instantiations.** Our CE block can be integrated into various advanced architectures, such as ResNet, VGGNet, ShuffleNet V2 and MobileNet V2, by inserting it in 'norm+ReLU-like' building block. The CE block is described in Fig.2(b). As discussed earlier, CE processes incoming features after the normalization layer by combining two branches, i.e. batch decorrelation (BD) and adaptive instance inverse (AII). Compared with SE block in Fig.2(a), the proposed CE block combines both instance and batch statistics, and it can consequently model dependencies among channels better.

A series of CENets can be constructed by integrating CE block into various advanced CNN architectures. For example, we consider the residual networks (ResNet). The core unit of the ResNet is the residual block that consists of '$1 \times 1$', '$3 \times 3$' and '$1 \times 1$' convolution layers, sequentially. Since the CE block is expected to help channel equalization, it would benefit from larger number of channels. Therefore, we employ CE block in the last '$1 \times 1$' convolution layer by plugging the CE module before ReLU non-linearity, as shown in Fig.2(c). As for CE-MobileNet V2, since the last '$1 \times 1$' convolution layer in the bottleneck is not followed by a ReLU activation, we insert CE in the '$3 \times 3$' convolution layer that also has a largest number of channels in the bottleneck. Following similar strategies, CE is further integrated into ShuffleNet V2 to construct CE-ShuffleNet V2. We provide extensive experiments evaluating all these CENets in Sec.4 and computational details in training and inference in Sec.F of Appendix.

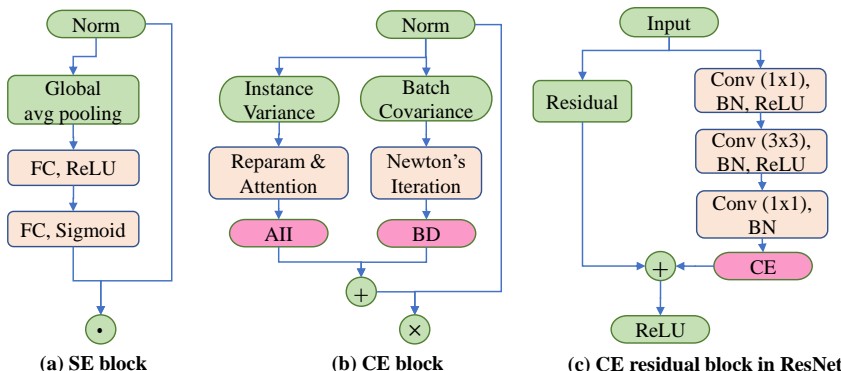

**(a) SE block**        **(b) CE block**        **(c) CE residual block in ResNet**

Figure 2: Illustrations of SE block (Hu et al., 2018) (a), CE block (b) and CE residual block in ResNet (c). $\odot$ denotes broadcast element-wise multiplication, $\oplus$ denotes broadcast elementwise addition and $\otimes$ denotes matrix multiplication. (b) shows CE has two lightweight branches, BN and AII. (c) shows CE can be easily stacked into many advanced networks such as ResNet with merely small extra computation.

**Equivalent $\gamma$.** Here we show how BD and AII benefit from each other through parameter $\gamma$. First, we disclose the mechanism in preventing inhibited channels behind the BD branch. Previous work (He et al., 2017b; Yu et al., 2018; Frankle & Carbin, 2018) revealed that $\gamma$ in BN can be used to prune less unimportant channels, implying that the representational power of feature map largely depends on the magnitude of $\gamma$. Combining Eqn.(4) and Eqn.(5), the output of BD can be expressed as $p_{nij}^{\text{BD}} = \text{Diag}(\Sigma^{-\frac{1}{2}}\gamma)\bar{x}_{nij} + \Sigma^{-\frac{1}{2}}\beta$. Comparing with Eqn.(2), an equivalent $\gamma$ can be defined as $\hat{\gamma} = \Sigma^{-\frac{1}{2}}\gamma$ for BD branch. The proposition 1 shows that BD explicitly increases the magnitude of $\hat{\gamma}$ in a feed-forward way, encouraging all channels to contribute to the feature learning process. The representational power is thus boosted in a channel basis. We provide the proof of proposition 1 in Sec.C of Appendix.

Furthermore, the original $\gamma$ in BN is also implicitly enlarged. It can be seen from Eqn.(6), a sufficient small $\gamma_c$ can cause degradation of the covariance matrix and then the convergence of Newton's iteration (Bini et al., 2005) cannot be guaranteed. As a result, once the network converges, $\gamma_c$ is not supposed to degrade. This will in turn bring many benefits to AII branch. Eqn.(8) shows that the input of AII is proportional to $\gamma$, meaning that the features fed into AII branch are enlarged as $\gamma$ increases. In this way, a bottleneck architecture in AII can learn more compact global information and model channel dependencies better.

**Proposition 1.** *Let $\Sigma$ be covariance matrix of feature maps after batch normalization. Assume that $\Sigma_k = \Sigma^{-\frac{1}{2}}, \forall k = 2, 3, \cdots, T$, then $\|\hat{\gamma}\|_1 > \|\gamma\|_1$. Especially, we have $|\hat{\gamma_i}| > |\gamma_i|$.*

**Connection with Nash Equilibrium.** We show an interesting connection between the proposed CE block and the well-known Nash Equilibrium in game theory (Leshem & Zehavi, 2009). To be specific, we bring novel insights on normalization from an optimization perspective. Suppose each channel obtains its output by maximizing capacity available to itself under some constraints. Especially, we restrict that each channel has a maximum budget and all the outputs are non-negative. Further, if we consider dependencies among channels, the channels are thought to play a non-cooperative game, named Gaussian interference game which admits a unique Nash Equilibrium solution (Laufer et al., 2006). In Sec.D of Appendix, we present the detailed construction of Gaussian interference game in the context of CNNs. It is worth noting that when all the outputs are activated (larger than 0), this Nash Equilibrium solution has an explicit expression. Under some mild approximations, it can be shown that the explicit Nash Equilibrium solution can surprisingly match the representation of CE in Eqn.(5). It shows that decorrelating features after normalization layer can be connected with Nash Equilibrium, implying that the proposed CE block indeed encourages every channel to contribute to the network's computation. We present detailed explanations about the connection between CE and Nash Equilibrium in Sec.D of Appendix.

|  | ResNet18 | | | ResNet50 | | | ResNet101 | | |
|---|---|---|---|---|---|---|---|---|---|
|  | Baseline | SE | CE | Baseline | SE | CE | Baseline | SE | CE |
| Top-1 | 70.4 | 71.4 | **71.9** | 76.6 | 77.6 | **78.3** | 78.0 | 78.5 | **79.0** |
| Top-5 | 89.4 | 90.4 | **90.8** | 93.0 | 93.7 | **94.1** | 94.1 | 94.1 | **94.6** |
| GFLOPs | 1.82 | 1.82 | 1.83 | 4.14 | 4.15 | 4.16 | 7.87 | 7.88 | 7.89 |
| CPU (s) | 3.69 | 3.69 | 4.13 | 8.61 | 11.08 | 11.06 | 15.58 | 19.34 | 17.05 |
| GPU (s) | 0.003 | 0.005 | 0.006 | 0.005 | 0.010 | 0.009 | 0.011 | 0.040 | 0.015 |

Table 1: Comparisons with baseline and SENet on ResNet-18, -50, and -101 in terms of accuracy, GFLOPs, CPU and GPU inference time on ImageNet. The top-1,-5 accuracy of our CE-ResNet is higher than SE-ResNet while the computational cost in terms of GFLOPs, GPU and CPU inference time remain nearly the same.

|  | MobileNet V2 $1\times$ | | | ShuffleNet V2 $0.5\times$ | | | ShuffleNet V2 $1\times$ | | |
|---|---|---|---|---|---|---|---|---|---|
|  | Top-1 | Top-5 | GFLOPs | Top-1 | Top-5 | GFLOPs | Top-1 | Top-5 | GFLOPs |
| Baseline | 72.5 | 90.8 | 0.33 | 59.2 | 82.0 | 0.05 | 69.0 | 88.6 | 0.15 |
| SE | 73.5 | 91.7 | 0.33 | 60.2 | 82.4 | 0.05 | 70.7 | 89.6 | 0.15 |
| CE | **74.6** | **91.7** | 0.33 | **60.5** | **82.7** | 0.05 | **71.2** | **89.8** | 0.16 |

Table 2: Comparisons with baseline and SE on lightweight networks, MobileNet V2 and ShuffleNet V2, in terms of accuracy and GFLOPs on ImageNet. Our CENet improves the top-1 accuracy by a large margin compared with SENet with nearly the same GFLOPs.

## 4 EXPERIMENTS

We evaluate our methods on two basic vision tasks, image classification on ImageNet and object detection/segmentation on COCO, where we demonstrate the effectiveness of the CE block.

### 4.1 IMAGE CLASSIFICATION ON IMAGENET

We first evaluate CE on the ImageNet benchmark. The training details are illustrated in Sec.F of Appendix.

**Performance comparison on ResNet.** We evaluate on representative residual network structures including ResNet18, ResNet50 and ResNet101. The CE-ResNet is then compared with baseline (plain ResNet) and SE-ResNet. For fair comparisons, we use publicly available code and re-implement baseline models and SE modules with their respective best settings in a unified Pytorch framework. To save computation, the CE blocks are selectively inserted into the last normalization layer of each residual block. Specifically, for ResNet18, we plug the CE block into each residual block. For ResNet50, CE is inserted into all residual blocks except for those layers with 2048 channels. For ResNet101, the CE blocks are employed in the first seven residual blocks.

As shown in Table 1, our proposed CE outperforms the BN baseline and SE block by a large margin with little increase of GFLOPs. Concretely, CE-ResNet18, CE-ResNet50 and CE-ResNet101 obtain top-1 accuracy increase of $1.5\%$, $1.7\%$ and $1.0\%$ compared with the corresponding plain ResNet architectures. The CE-ResNet50 even outperforms the plain ResNet101 (78.0). We plot training and validation loss during the training process for ResNet50, SE-ResNet50 and CE-ResNet50 in Sec.E of Appendix.

We also analyze the complexity of BN, SE, and CE in terms of GFLOPs, GPU and CPU running time. We evaluate the inference time[1] with a mini-batch of 32. In term of GFLOPs, the CE-ResNet18, CE-ResNet50, CE-ResNet101 has only $0.242\%$ and $0.241\%$ relative increase in GFLOPs compared with plain ResNet. Additionally, the CPU and GPU inference time of CENet is nearly the same with SENet.

**Performance comparison on light-weight networks**. We further verify the effectiveness of our proposed CE in two representative light-weight networks, MobileNet V2 and ShuffleNet V2. The results of comparison are listed in Table 2. It is seen that CE blocks bring conspicuous improvements

---

[1] The CPU type is Intel Xeon CPU E5-2682 v4, and the GPU is NVIDIA GTX1080TI. The implementation is based on Pytorch

|  | BN | | GN | | IN | | LN | |
|---|---|---|---|---|---|---|---|---|
|  | top-1 | top-5 | top-1 | top-5 | top-1 | top-5 | top-1 | top-5 |
| Baseline | 76.6 | 93.0 | 75.6 | 92.8 | 74.2 | 91.9 | 71.6 | 89.9 |
| Baseline+CE | **78.3** | **94.1** | **76.2** | **92.9** | **76.0** | **92.7** | **73.3** | **91.3** |
| Increase | +1.7 | +1.1 | +0.6 | +0.1 | +1.8 | +0.8 | +1.7 | +1.4 |

Table 3: CE improves top-1 and top-5 accuracy of various normalization methods on ImageNet with ResNet50 as backbone.

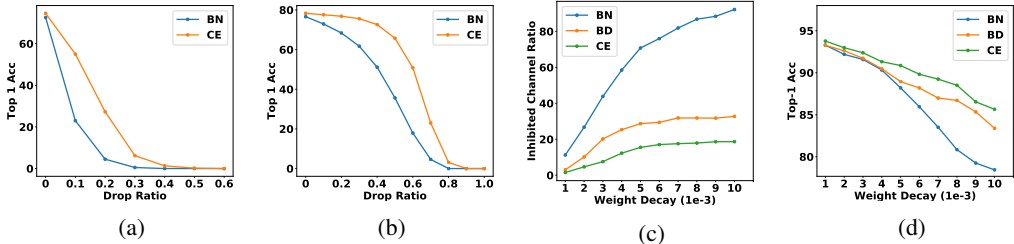

(a)        (b)        (c)        (d)

Figure 3: (a) & (b) show channel drop ratios versus top-1 accuracy for MobileNet V2 and ResNet50 on ImageNet dataset respectively. We randomly ablate channels with an increasing fraction in the first normalization layers. Note that CE-ResNet50 and CE-MobileNet V2 are more robust to cumulative ablation of channels than those networks trained with only BN, suggesting that CE also helps channels equalization on ImageNet. For (c) and (d), we train VGGNet in CIFAR-10 under different weight decays. It is observed that the networks trained with the proposed BD and CE were consistently and substantially more robust to the increasing strength of weight decay than those trained with single BN.

in performance at a minimal increase in computational burden on mobile settings. For MobileNet V2, we see that CE blocks even improves top-1 accuracy of baseline by 2.1%.

**Other Normalizers**. In addition to BN, CE is also effective for other normalization technologies, since inhibited channels emerges in many well-known normalizers as shown in Fig.1. To prove this, we conduct experiments using ResNet-50 under different normalizers including batch normalization (BN), group normalization (GN), instance normalization (IN), and layer normalization (LN). For these experiments, we stack CE block after the above normalizers to see whether CE helps other normalization methods. As shown in Table 3, our CE generalize well over different normalization technology, improving the performance by 0.6-1.8 top-1 accuracy.

### 4.2 ANALYSIS OF CE

In this section, we first demonstrate that CE is able to equalize the importance of all channels and then analyze the effects of BD and AII branches separately on CIFAR10 and ImageNet datasets. More discussions about CE are provided in Sec.E of Appendix.

**CE is effective in channel equalization.** In Fig.1, we have demonstrated that CE is able to alleviate inhibited channels, which is a necessary condition of channel equalization. Here, we further show that the ability that CE can prevent inhibited channels is robust to a wide range of strength of weight decay. As shown in Fig.3(c,d), CE prevents inhibited channels and retains higher performance under different strengths of weight decay.

Next, we verify whether CE can help channel equalization by an ablation approach used in Morcos et al. (2018). Typically, the importance of a single channel to the network's computation can be measured by the relative performance drop once that channel is removed (clamping activity a feature map to zero). In this regard, the more reliant a network is on a small set of channels, the more quickly the accuracy will drop if those channels are ablated. On the contrary, if the importance of channels to the network's computation are more equal, the accuracy will drop more gently. With this powerful technique, we see how ResNet50 and MobileNet V2 with CE blocks respond to cumulative random ablation of channels. We plot the ablation ratio versus the top-1 accuracy in Fig.3(a,b). As we can see, our CE block is able to resist the cumulative random ablation of channels on both ResNet50 and MobileNet V2, showing that CE can effectively equalize the importance of channels. For example, the top-1 accuracy of our CE-ResNet50 is 1.7 higher

| Method | Plain ResNet50 | BD-ResNet50 | AII-ResNet50 | CE-ResNet50 |
|--------|----------------|-------------|--------------|-------------|
| top-1 | 76.6 | 77.0 (+0.4) | 77.3 (+0.7) | **78.3 (+1.7)** |

Table 4: Results of batch covariance decorrelation, adaptive variance inverse and channel equilibrium. We use ResNet-50 as the basic structure. The top-1 accuracy increase (1.7) of CE-ResNet is higher than combined top-1 accuracy increase (1.1) of BD-ResNet and AII-ResNet, indicating the effects of BD and AII branch is complementary.

than the original ResNet50 if no channels are ablated, but when 70% channels are ablated, CE-ResNet50 still obtain 23.0 top-1 accuracy, while the original ResNet50 gets only 4.6 top-1 accuracy.

**BD is able to mitigate inhibited channels.** As proved in Proposition 1, equivalent $\gamma$ in the BD branch is explicitly enlarged, leading to the expansion of representational power of all channels. Here we investigate this property experimentally with a single BD branch. The inhibited channels ratio are measured by the percentage of feature maps whose values are less than 1e-2. Fig.3(d) shows inhibited channels ratio under a wide range of weight decay for BN, BD and CE. It is observed that the top-1 accuracy of VGGNet with BN drops significantly as the weight decay increases, but BD can reduce accuracy drop. For example, when the weight decay is 1e-3, the top-1 accuracy of BD is only 0.1 higher than BN, but when the weight decay reaches to 1e-2, BD is 4.95 higher. Moreover, the inhibited channels ratio of CE is even lower than BD while the top-1 accuracy is higher, which can demonstrate that CE can strengthen the effect of alleviation of inhibited channels compared with single BD.

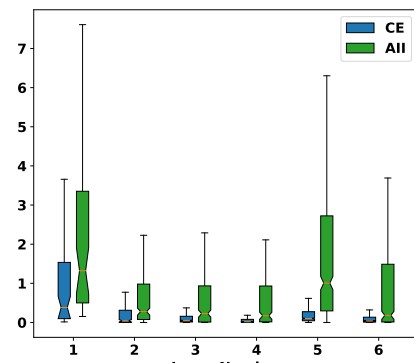

Figure 4: We do principal component analysis (PCA) on the input of AII sub-network, the variance of each channel. This figure show the box chart of principal components. CENet has lower means and variances than AIINet, indicating the input of AII sub-network in CENet is more equal and informative.

**AII helps CE learn preciser feature representation.** First, as discussed in Sec.3.3, AII benefits from BD such that the features fed into AII branch are more informative. To see this, we train ResNet50 with a single AII or CE branch, termed AII-ResNet50 or CE-ResNet50. We do principal component analysis (PCA) on the inputs of AII branch in AII-ResNet50 and the counterpart in CE-ResNet50 and plot the box chart of principal components. As is shown in Fig.4, the input of AII branch in CE-ResNet50 gets much lower means and variances, meaning that the input feature has more valid basis and thus more informative.From the Grad-CAM visualization provided in Sec.E of Appendix, we find that AII helps CE learn preciser feature representation.

**BD and AII are complementary**. Here, we verify that BD and AII are complementary to each other. We train plain ResNet50, BD-ResNet50, AII-ResNet50, and CE-ResNet50 for comparison. The top-1 accuracy is reported in Table 4. It is observed that the BD-ResNet50 and AII-ResNet50 are 0.4 and 0.7 higher than the plain ResNet-50 respectively. However, when they are combined, the top-1 accuracy improves by 1.7, higher than combined accuracy increase (1.1), which demonstrates that they benefit from each other.

## 4.3 OBJECT DETECTION AND INSTANCE SEGMENTATION ON COCO

We assess the generalization of our CE block on detection/segmentation track using the COCO2017 dataset ( Lin et al. (2014)). We train our model on the union of 80k training images and 35k validation images and report the performance on the mini-val 5k images. Mask-RCNN is used as the base detection/segmentation framework. The standard COCO metrics of Average Precision (AP) for bounding box detection (APbb) and instance segmentation (APm) is used to evaluate our methods. In addition, we adopt two common training settings for our models, (1) freezing the vanilla batch normalization and channel equilibrium layer and (2) updating parameters with the synchronized version. For vanilla BN and CE layers, all the gamma, beta parameters, and the tracked running

| Backbone | $AP^b$ | $AP^b_{.5}$ | $AP^b_{.75}$ | $AP^m$ | $AP^m_{.5}$ | $AP^m_{.75}$ |
|---|---|---|---|---|---|---|
| ResNet50 | 38.6 | 59.5 | 41.9 | 34.2 | 56.2 | 36.1 |
| CE-ResNet50 | 40.8 | **62.7** | 44.3 | 36.9 | 59.2 | 39.4 |
| SyncCE-ResNet50 | **42.0** | 62.6 | **46.1** | **37.5** | **59.5** | **40.3** |
| ResNet101 | 40.3 | 61.5 | 44.1 | 36.5 | 58.1 | 39.1 |
| CE-ResNet101 | **41.6** | **62.8** | **45.8** | **37.4** | **59.4** | **40.0** |

Table 5: Detection and segmentation results in COCO using Mask-RCNN We use the pretrained CE-ResNet50 model (78.3) and CE-ResNet101 (79.0) in ImageNet to train our model. CENet can consistently improve both box AP and segmentation AP by a large margin.

statistics are frozen. In contrast, for the synchronized version, the running mean and variance for batch normalization and the covariance for CE layers are computed across multiple GPUs. The gamma and beta parameters are updated during training while $\tilde{F}$ and $\lambda$ are frozen to prevent overfitting. We use MMDetection training framework with ResNet50/ResNet101 as basic backbones and all the hyper-parameters are the same as Chen et al. (2019). Table 5 shows the detection and segmentation results. The results show that compared with vanilla BN, our CE block can consistently improve the performance. For example, our fine-tuned CE-ResNet50 is 2.2 AP higher in detection and 2.7 AP higher in segmentation. For the sync BD version, CE-ResNet50 gets **42.0** AP in detection and **37.5** AP in segmentation, which is the best performance for ResNet50 to the best of our knowledge. To sum up, these experiments demonstrate the generalization ability of CE blocks in other tasks.

## 5 CONCLUSION

In this paper, we presented a novel network block, termed as Channel Equilibrium (CE). The CE block conditionally decorrelates feature maps after normalization layer by switching between batch decorrelation branch and adaptive instance inverse branch. We show that CE is able to explicitly alleviate the inhibited channels and help channel equalization, enhancing the representational power of a neural network in a feed-forward way. Specifically, CE can be stacked between the normalization layer and the ReLU function, making it flexible to be integrated into many advanced CNN architectures. The superiority of CE blocks has been demonstrated on the task of image classification and instance segmentation. We hope that the analysis of channel equalization in CE could bring a new perspective for future work in architecture design.

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

# Appendix

## A   MORE RELATED WORK

**Sparsity in ReLU.** An attractive property of ReLU (Sun et al., 2015; Nair & Hinton, 2010) is sparsity, which brings potential advantages such as information disentangling and linear separability. However, Lu et al. (2019) and Mehta et al. (2019) pointed out that some ReLU neurons may become inactive and output 0 values for any input. Previous work tackled this issue by designing new activation functions, such as ELU (Clevert et al., 2015) and Leaky ReLU (Maas et al., 2013). Recently, Lu et al. (2019) also tried to solve this problem by modifying initialization scheme. Different from these work, we focus on explicitly preventing inhibited channel in a feed-forward way by the proposed CE blocks.

**Normalization and decorrelation.** There are many practices on normalizer development, such as Batch Normalization (BN) (Ioffe & Szegedy, 2015), Group normalization (GN) (Wu & He, 2018) and Switchable Normalization (Luo et al., 2018). A normalization scheme is typically applied after a convolution layer and contains two stages: standardization and rescaling. Another type of normalization methods not only standardizes but also decorrelates features, like DBN (Huang et al., 2018), IterNorm (Huang et al., 2019) and switchable whitening (Pan et al., 2019). Despite their success in performance improvement, little is explored about relation between those methods and inhibited channels. Fig.1 shows that inhibited channels emerges in VGGNet where 'BN+ReLU' or 'LN+ReLU' are used. Unlike previous decorrelated normalizations where decorrelation operation is applied after a convolution layer, our CE explicitly decorrelates features after normalization.

## B   COMPUTATION DETAILS IN 'BN-CE-ReLU' BLOCK

As discussed before, CE processes incoming features after normalization layer by combining two branches, i.e. batch decorrelation and adaptive instance inverse. The former computes a covariance matrix and the latter calculates instance variance. We now take 'BN-CE-ReLU' block as an example to show the computation details of statistics in it. Given a tensor $x \in \mathbb{R}^{N \times C \times H \times W}$, the mean and variance in IN (Ulyanov et al., 2016) are calculated as:

$$\mu_{\mathrm{IN}}^{nc} = \frac{1}{HW} \sum_{i,j}^{H,W} x_{ncij}, \quad (\sigma_{\mathrm{IN}}^2)^{nc} = \frac{1}{HW} \sum_{i,j}^{H,W} (x_{ncij} - \mu_{\mathrm{IN}}^{nc})^2 \tag{11}$$

Hence, we have $\mu_{\mathrm{IN}}, \sigma_{\mathrm{IN}}^2 \in \mathbb{R}^{N \times C}$. Then, the statistics in BN can be reformulated as follows:

$$\mu_{\mathrm{BN}}^c = \frac{1}{NHW} \sum_{n,i,j}^{N,H,W} x_{ncij} = \frac{1}{N} \sum_i^N \frac{1}{HW} \sum_{i,j}^{H,W} x_{ncij}$$

$$(\sigma_{\mathrm{BN}}^2)^c = \frac{1}{NHW} \sum_{n,i,j}^{N,H,W} (x_{ncij} - \mu_{\mathrm{BN}}^c)^2$$

$$= \frac{1}{N} \sum_n^N \frac{1}{HW} \sum_{i,j}^{H,W} (x_{ncij} - \mu_{\mathrm{IN}}^{nc} + \mu_{\mathrm{IN}}^{nc} - \mu_{\mathrm{BN}}^c)^2 \tag{12}$$

$$= \frac{1}{N} \sum_n^N (\frac{1}{HW} \sum_{i,j}^{H,W} (x_{ncij} - \mu_{\mathrm{IN}}^{nc})^2 + (\mu_{\mathrm{IN}}^{nc} - \mu_{\mathrm{BN}}^c)^2)$$

$$= \frac{1}{N} \sum_n^N (\sigma_{\mathrm{IN}}^2)^{nc} + \frac{1}{N} \sum_n^N (\mu_{\mathrm{IN}}^{nc} - \mu_{\mathrm{BN}}^c)^2$$

Then, we have $\mu_{\mathrm{BN}} = \mathbb{E}[\mu_{\mathrm{IN}}]$ and $\sigma_{\mathrm{BN}}^2 = \mathbb{E}[\sigma_{\mathrm{IN}}^2] + \mathrm{D}[\mu_{\mathrm{IN}}]$, where $\mathbb{E}[\cdot]$ and $\mathrm{D}[\cdot]$ denote expectation and variance operators over N samples. Further, the input of AII is instance variance of features

after BN, which can be calculated as follows:

$$\sigma_{nc}^2 = \frac{1}{HW} \sum_{i,j}^{H,W} \left[ (\gamma_c \frac{x_{ncij} - \mu_{\text{BN}}^c}{\sigma_{\text{BN}}^c} + \beta_c) - (\gamma_c \frac{\mu_{\text{IN}}^{nc} - \mu_{\text{BN}}^c}{\sigma_{\text{BN}}^c} + \beta_c) \right]^2$$

$$= \frac{\gamma_c^2}{(\sigma_{\text{BN}}^2)^c} \frac{1}{HW} \sum_{i,j}^{H,W} (x_{ncij} - \mu_{\text{IN}}^{nc})^2 \tag{13}$$

$$= \frac{\gamma_c^2 (\sigma_{\text{IN}}^2)^{nc}}{(\sigma_{\text{BN}}^2)^c}$$

At last, the output of BN is $\tilde{x}_{ncij} = \gamma_c \bar{x}_{ncij} + \beta_c$, then the entry in c-th row and d-th column of covariance matrix $\Sigma$ of $\tilde{x}$ is calculated as follows:

$$\Sigma_{cd} = \frac{1}{NHW} \sum_{n,i,j}^{N,H,W} (\gamma_c \bar{x}_{ncij})(\gamma_d \bar{x}_{ndij}) = \gamma_c \gamma_d \rho_{cd} \tag{14}$$

where $\rho_{cd}$ is the element in c-th row and j-th column of correlation matrix of $\bar{x}$. Thus, we can write $\Sigma$ into vector form: $\Sigma = \gamma\gamma^{\mathsf{T}} \odot \frac{1}{M} \bar{x}\bar{x}^{\mathsf{T}}$ if we reshape $\tilde{x}$ to $\tilde{x} \in \mathbb{R}^{C \times M}$ and $M = N \cdot H \cdot W$.

## C  PROOF OF PROPOSITION 1

**Proposition 1.** *Let $\Sigma$ be covariance matrix of feature maps after batch normalization. Assume that $\Sigma_k = \Sigma^{-\frac{1}{2}}$, $\forall k = 2, 3, \cdots, T$, then $\|\hat{\gamma}\|_1 > \|\gamma\|_1$. Especially, we have $|\hat{\gamma}_i| > |\gamma_i|$*

Proof. Since $\Sigma_k = \Sigma^{-\frac{1}{2}}$, $\forall k = 2, 3, \cdots, T$, we have $\Sigma_k \gamma = \frac{1}{2} \Sigma_{k-1} (3I - \Sigma_{k-1}^2 \Sigma)\gamma = \Sigma_{k-1}\gamma$. Therefore, we only need to show $\|\hat{\gamma}\|_1 = \|\Sigma_T \gamma\|_1 = \cdots = \|\Sigma_2 \gamma\|_1 > \|\gamma\|_1$. Now, we show that for $k = 2$ we have $\left\| \frac{1}{2}(3I - \Sigma)\gamma \right\|_1 > \|\gamma\|_1$. From Eqn.(6), we know that $\Sigma = \frac{\gamma\gamma^{\mathsf{T}}}{\|\gamma\|_2^2} \odot \rho$ where $\rho$ is the correlation matrix of $\tilde{x}$ and $-1 \leq \rho_{ij} \leq 1$, $\forall i, j \in [C]$. Then, we have

$$\frac{1}{2}(3I - \Sigma)\gamma = \frac{1}{2}(3I - \frac{\gamma\gamma^{\mathsf{T}}}{\|\gamma\|_2^2} \odot \rho)\gamma$$

$$= \frac{1}{2}(3\gamma - (\frac{\gamma\gamma^{\mathsf{T}}}{\|\gamma\|_2^2} \odot \rho)\gamma)$$

$$= \frac{1}{2}(3\gamma - \frac{1}{\|\gamma\|_2^2} \left[ \sum_j^C \gamma_1 \gamma_j \rho_{1j} \gamma_j, \sum_j^C \gamma_2 \gamma_j \rho_{2j} \gamma_j, \cdots, \sum_j^C \gamma_C \gamma_j \rho_{Cj} \gamma_j \right]^{\mathsf{T}})$$

$$= \frac{1}{2}(3\gamma - \frac{1}{\|\gamma\|_2^2} \left[ \sum_j^C \gamma_1 \gamma_j \rho_{1j} \gamma_j, \sum_j^C \gamma_2 \gamma_j \rho_{2j} \gamma_j, \cdots, \sum_j^C \gamma_C \gamma_j \rho_{Cj} \gamma_j \right]^{\mathsf{T}}) \tag{15}$$

$$= \frac{1}{2} \left[ (3 - \sum_j^C \frac{\gamma_j^2 \rho_{1j}}{\|\gamma\|_2^2})\gamma_1, (3 - \sum_j^C \frac{\gamma_j^2 \rho_{2j}}{\|\gamma\|_2^2})\gamma_2, \cdots, (3 - \sum_j^C \frac{\gamma_j^2 \rho_{Cj}}{\|\gamma\|_2^2})\gamma_C \right]^{\mathsf{T}}$$

Note that $|3 - \sum_j^C \frac{\gamma_j^2 \rho_{ij}}{\|\gamma\|_2^2}| \geq 3 - |\sum_j^C \frac{\gamma_j^2 \rho_{ij}}{\|\gamma\|_2^2}| \geq 3 - \sum_j^C \frac{\gamma_j^2}{\|\gamma\|_2^2} = 2$, where the last equality holds iff $\rho_{ij} = 1$, $\forall i, j \in [C]$. However, this is not the case in practice. Hence we have

$$\left| \left[ \frac{1}{2}(3I - \Sigma)\gamma \right]_i \right| = \left| \frac{1}{2}(3 - \sum_j^C \frac{\gamma_j^2 \rho_{ij}}{\|\gamma\|_2^2})\gamma_i \right| > |\gamma_i| \tag{16}$$

Therefore, we have $\|\hat{\gamma}\|_1 > \|\gamma\|_1$. Here completes the proof.

# D   CONNECTION BETWEEN CE BLOCK AND NASH EQUILIBRIUM

We first introduce the definition of Gaussian interference game in context of CNN and then build the connection between a CE block and Nash Equilibrium. For clarity of notation, we omit the subscript $n$ for a concrete sample.

Let the channels $1, 2, \cdots, C$ operate over $H \times W$ pixels. Assume that the C channels have dependencies $G = \{g_{cd}(i,j)\}_{c,d=1}^{C,C}$. Each pixel is characterized by a power gain $h_{cij} \geq 0$ and channel noise strength $\sigma_c > 0$. In context of normalization, we suppose $h_{cij} = \bar{x}_{cij} + \delta$ where $\bar{x}_{cij}$ is standardized pixel in Eqn.(2) and $\delta$ is sufficiently large to guarantee a non-negative power gain. Assume that c-th channel is allowed to transmit a total power of $P_c$ and we have $\sum_{i,j=1}^{H,W} p_{cij} = P_c$. Besides, each channel can transmit a power vector $p_c = (p_{c11}, \cdots, p_{cHW})$. Since normalization layer is often followed by a ReLU activation, we restrict $p_{cij} \geq 0$. What we want to maximize the capacity transmitted over the c-th channel, $\forall c \in [C]$, then the maximization problem is given by:

$$\max \ C_c(p_1, p_2, \cdots, p_C) = \sum_{i,j=1}^{h,W} \ln \left( 1 + \frac{g_{cc}p_{cij}}{\sum_{d \neq c} g_{cd}p_{dij} + \sigma_c/h_{cij}} \right)$$

$$s.t. \quad \begin{cases} \sum_{i,j=1}^{H,W} p_{cij} = P_c, \\ p_{cij} \geq 0, \end{cases} \quad \forall i \in [H], j \in [W] \tag{17}$$

where $C_c$ is the capacity available to the c-th channel given power distributions $p_1, p_2, \cdots, p_C$. In game theory, C channels and solution space of $\{p_{cij}\}_{c,i,j=1}^{C,H,W}$ together with pay-off vector $\mathbf{C} = (C_1, C_2, \cdots, C_C)$ form a Gaussian interference game $\mathbb{G}$. Different from basic settings in $\mathbb{G}$, here we do not restrict dependencies $g_{cd}$ to $(0, 1)$. It is known that $\mathbb{G}$ has a unique Nash Equilibrium point whose definition is given as below,

**Definition 1.** *An C-tuple of strategies $(p_1, p_2, \cdots, p_C)$ for channels $1, 2, \cdots, C$ respectively is called a Nash equilibrium iff for all c and for all p (p a strategy for channel c)*

$$C_c(p_1, \cdots, p_{c-1}, p, p_{c+1}, \cdots, p_C) \leq C_c(p_1, p_2, \cdots, p_C) \tag{18}$$

i.e., given that all other channels $d \neq c$ use strategies $p_d$, channel c best response is $p_c$. Since $C_1, C_2, \cdots, C_C$ are concave in $p_1, p_2, \cdots, p_C$ respectively, KKT conditions imply the following theorem.

**Theorem 1.** *Given pay-off in Eqn.(17), $(p_1^*, \cdots, p_C^*)$ is a Nash equilibrium point if and only if there exist $v_0 = (v_0^1, \cdots, v_0^C)$ (Lagrange multiplier) such that for all $i \in [H]$ and $j \in [W]$,*

$$\frac{g_{cc}}{\sum_d g_{cd}p_{dij}^* + \sigma_c/h_{cij}} \begin{cases} = v_0^c \text{ for } p_{cij}^* > 0 \\ \leq v_0^c \text{ for } p_{cij}^* = 0 \end{cases} \tag{19}$$

Proof. The Lagrangian corresponding to minimization of $-C_c$ subject to the equality constraint and non-negative constraints on $p_{cij}$ is given by

$$L_c = -\sum_{i,j=1}^{h,W} \ln \left( 1 + \frac{g_{cc}p_{cij}}{\sum_{d \neq c} g_{cd}p_{dij} + \sigma_c/h_{cij}} \right) + v_0^c(\sum_{i,j=1}^{H,W} p_{cij} - P_c) + \sum_{i,j=1}^{H,W} v_1^{cij}(-p_{cij}). \tag{20}$$

Differentiating the Lagrangian with respect to $p_{cij}$ and equating the derivative to zero, we obtain

$$\frac{g_c c}{\sum_d g_{cd}p_{cij} + \sigma_c/h_{cij}} + v_1^{cij} = v_0^c \tag{21}$$

Now, using the complementary slackness condition $v_1^{cij}p_{cij} = 0$ and $v_1^{cij} \geq 0$, we obtain condition (19). This completes the proof.

By Theorem 1, the unique Nash Equilibrium point can be explicitly written as follows when $p_{cij}^* > 0$,

$$p_{ij}^* = G^{-1} \left( \text{Diag}(v_0)^{-1}\text{diag}(G) - \text{Diag}(h_{ij})^{-1}\sigma \right) \tag{22}$$

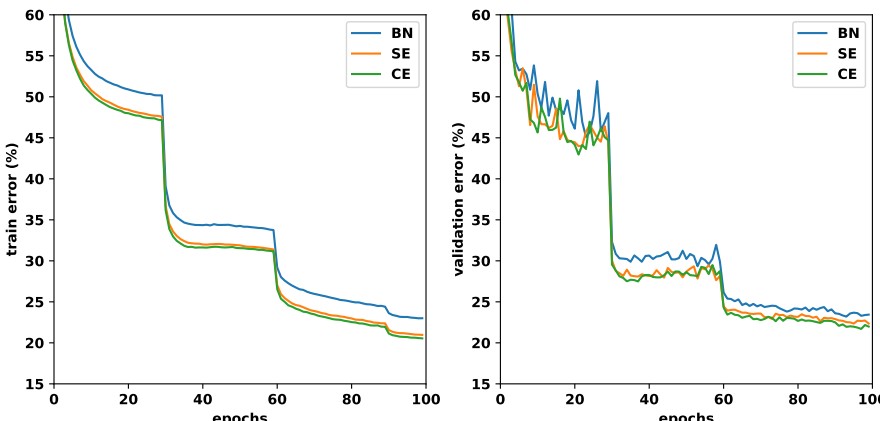

Figure 5: Training and validation error curves on ImageNet with ResNet50 as backbone for BN, SE and CE.

where $p_{ij}^*, h_{ij}, \sigma \in \mathbb{R}^C$ and $v_0 \in \mathbb{R}^C$ are Lagrangian multipliers corresponding to equality constraints. Note that a approximation can be made using Taylor expansion as follow: $-\frac{\sigma_c}{h_{cij}} = \sigma_c(2 + h_{cij} + \mathcal{O}((1 + h_{cij})^2))$. Thus, a linear proxy to Eqn.(22) can be written as

$$p_{ij}^* = G^{-1}\left(\text{Diag}(\sigma)\bar{x}_{ij} + \text{Diag}(v_0)^{-1}\text{diag}(G) + (2 + \delta)\sigma\right) \tag{23}$$

Let $G = [D_n]^{\frac{1}{2}}, \gamma = \sigma$ and $\beta = \text{Diag}(v_0)^{-1}\text{diag}(G) + (2 + \delta)\sigma$, Eqn.(23) can surprisingly match CE unit in Eqn.(5), implying that the proposed CE block indeed performs a mechanism on channel equalization. In Gaussian interference game, $\sigma$ is known and $v_0$ can be determined when budget $P_c$'s are given. However, $\gamma$ and $\beta$ are learned by SGD in deep neural networks.

## E    EXPERIMENTS

### E.1    TRAINING AND VALIDATION CURVES

We plot training and validation loss during the training process for ResNet50, SE-ResNet50 and CE-ResNet50 in Fig.5. We can observe that CE-ResNet50 consistently have lower training and validation errors over the whole training period, indicating that CE improves both learning capacity and generalization ability.

### E.2    MORE DISCUSSION ABOUT CE

As discussed in related work in Sec.A, many methods have been proposed to improve normalizers and ReLU activation. The ablation approach in Morcos et al. (2018) is used to see whether and how these methods help channel equalization. We demonstrate the effectiveness of CE by answering the following questions.

**Do other ReLU-like activation functions help channel equalization?** Two representative improvements on ReLU function, i.e. ELU (Clevert et al., 2015) and LReLU (Maas et al., 2013), are employed to see whether other ReLU-like activation functions can help channel equalization. We plot the cumulative ablation curve that depicts ablation ratio versus the top-1 accuracy on CIFAR10 dataset in Fig.6(a). The baseline curve is 'BN+ReLU'. As we can see, the top-1 accuracy curve of 'BN+LReLU' drops more gently, implying that LReLU helps channel equalization. But 'ELU+ReLU' has worse cumulative ablation curve than 'BN+ReLU'. By contrast, the proposed CE block improves the recognition performance of 'BN+ReLU' (higher top-1 accuracy) and promotes channel equalization most (the most gentle cumulative ablation curve).

**Do the adaptive normalizers help channel equalization?** We experiment on a representative adaptive normalization method (i.e. SN), to see whether it helps channel equalization. SN learns to

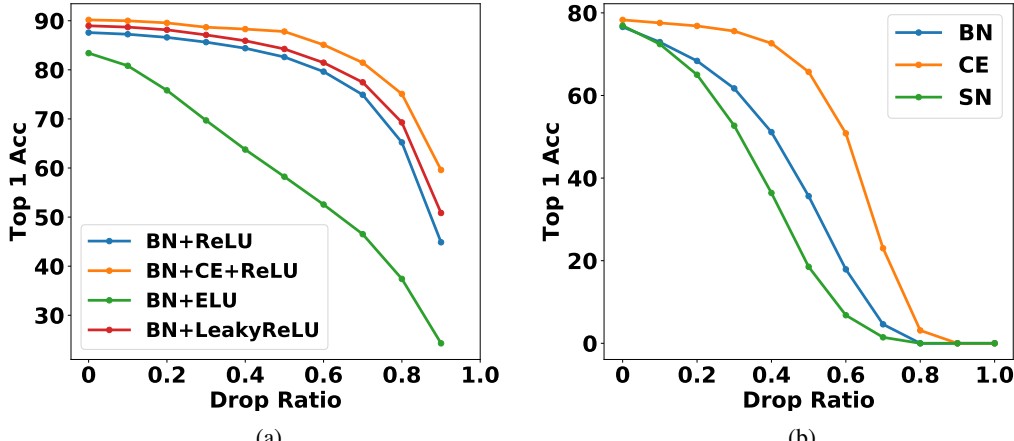

(a)            (b)

Figure 6: (a) compares the cumulative ablation curves of 'BN+ReLU', 'BN+ELU', 'BN+LReLU' and 'BN+CE+ReLU' with VGGNet on CIFAR-10 dataset. We see that the Both LReLU and CE can improve the channel equalization in 'BN+ReLU' block. (b) compares the cumulative ablation curves of 'BN+ReLU', 'SN+ReLU' and 'BN+CE+ReLU' with ResNet-50 on ImageNet dataset. The proposed CE consistently improves the channel equalization of 'BN+RelU' block. Note that 'BN+CE+ReLU' achieves the highest top-1 accuracy on both two datasets compared to its counterparts (when drop ration is 0).

|  | Top-1 acc |
| --- | --- |
| CE2-ResNet50 | 77.9 |
| CE3-ResNet50 | 78.3 |

Table 6: We add CE after the second (CE2-ResNet50) and third (CE3-ResNet50) batch normalization layer in each residual block. The channel of the third batch normalization is 4 times than that of the second one but the top-1 accuracy of CE3-ResNet50 outperforms CE2-ResNet50 by 0.4, which indicates CE benefits from larger number of channels.

select an appropriate normalizer from IN, BN and LN for each channel. The cumulative ablation curves are plotted on ImageNet dataset with ResNet-50 under blocks of 'BN+ReLU', 'SN+ReLU' and 'BN+CE+ReLU'. As shown in Fig.6(b), SN even does damage to channel equalization when it is used to replace BN. However, 'BN+CE+ReLU' shows the most gentle cumulative ablation curve, indicating the effectiveness of CE block in channel equalization. Compared with SN, ResNet-50 with CE block also achieves better top-1 accuracy (78.3 vs 76.9), showing that channel equalization is important for block design in a CNN.

**Integration strategy of CE block.** We put CE in different position of a bottleneck in ResNet50, which consists of three "Conv-BN-ReLU" basic blocks. The channel of the third block is 4 times than that of the second one. We compare the performance of CE-ResNet50 by putting CE in the second block (CE2-ResNet50) or the third block (CE3-ResNet50). As shown in Table 6, the top-1 accuracy of CE3-ResNet-50 outperforms CE2-ResNet50 by 0.4, which indicates that our CE block benefits from larger number of channels.

### E.3 GRAD-CAM VISUALIZATION

We claim that AII learns adaptive inverse of variance for each channel in a self-attention manner. Fed into more informative input, AII is expected to make the network respond to different inputs in a highly class-specific manner. In this way, it helps CE learn preciser feature representation. To verify this, we employ an off-the-shelf tool to visualize the class activation map (CAM) Selvaraju et al. (2017). We use ResNet50, BD-ResNet50, and CE-ResNet50 trained on ImageNet for comparison. As shown in Fig.7, the heat maps extracted from CAM for CE-ResNet50 have more coverage on the

(a) Origin  (b) ResNet50  (c) BD-ResNet50  (d) CE-ResNet50

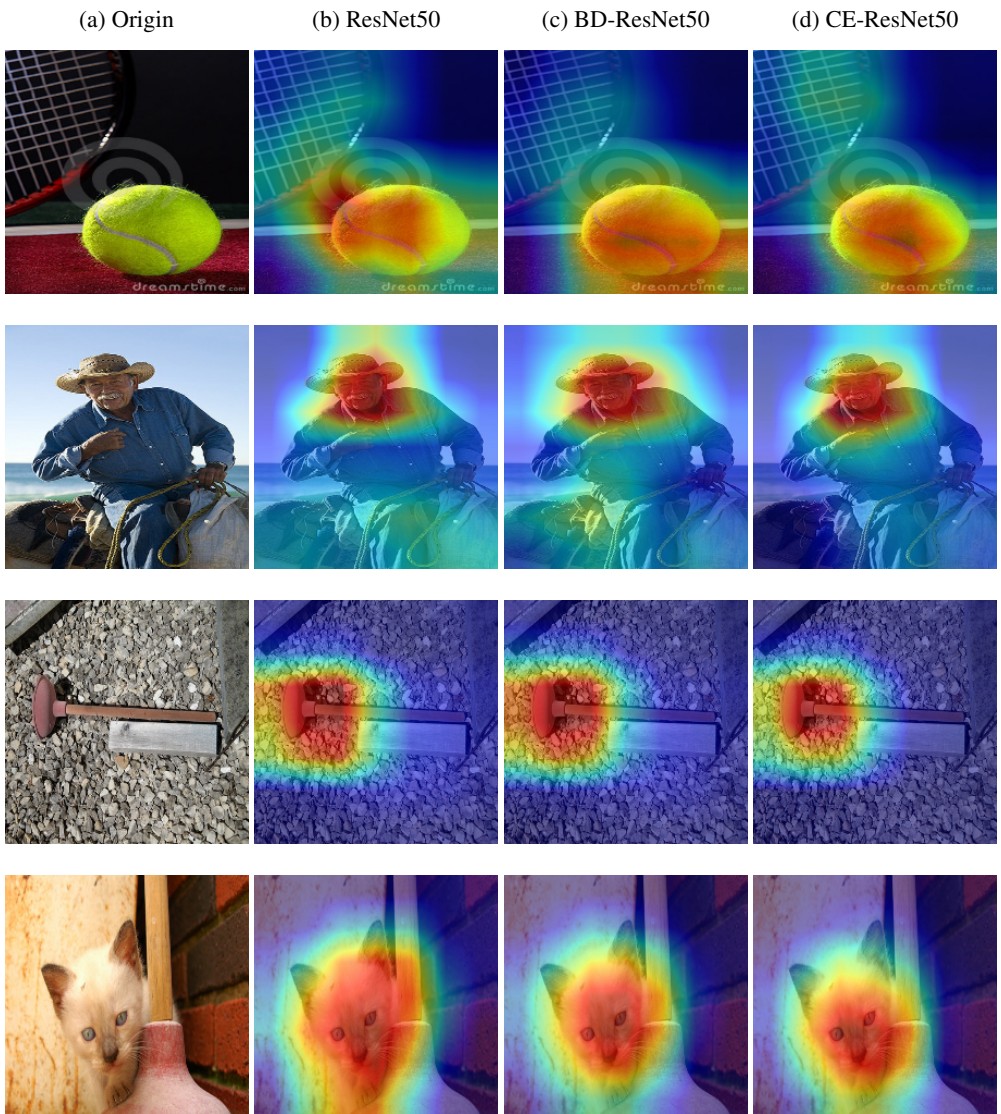

Figure 7: Grad-cam visualization results from the final convolutional layer for plain ResNet50, SE-ResNet50, and CE-ResNet50.

object region and less coverage on the background region. It shows that the AII branch helps CE learn preciser information from the images.

## F    TRAINING AND INFERENCE

**Moving average in inference**. Unlike previous methods in manual architecture design that do not depend on batch estimated statistics, the proposed CE block requires computing the inverse square root of a batch covariance matrix $\Sigma$ and a global variance scale $s$ in each training step. To make the output depend only on the input, deterministically in inference, we use the moving average to calculate the population estimate of $\hat{\Sigma}^{-\frac{1}{2}}$ and $\hat{s}^{-\frac{1}{2}}$ by following the below updating rules:

$$\hat{\Sigma}^{-\frac{1}{2}} = (1-m)\hat{\Sigma}^{-\frac{1}{2}} + m\Sigma^{-\frac{1}{2}}, \;\; \hat{s}^{-\frac{1}{2}} = (1-m)\hat{s}^{-\frac{1}{2}} + m \cdot s^{-\frac{1}{2}} \quad (24)$$

where $s$ and $\Sigma$ are the variance scale and covariance calculated within each mini-batch during training, and $m$ denotes the momentum of moving average. It is worth noting that since $\hat{\Sigma}^{-\frac{1}{2}}$ is fixed

during inference, the BD branch does not introduce extra costs in memory or computation except for a simple linear transformation ( $\hat{\Sigma}^{-\frac{1}{2}}\tilde{x}$ ).

**Model and computational complexity**. The main computation of our CE includes calculating the covariance and inverse square root of it in the BD branch and computing two FC layers in the AII branch. We see that there is a lot of space to reduce computational cost of CE. For BD branch, given an internal feature $x \in \mathbb{R}^{N \times C \times H \times W}$, the cost of calculating a covariance matrix is $2NHWC^2$, which is comparable to the cost of convolution operation. A pooling operation can be employed to downsample featuremap for too large $H$ and $W$. In this way, the complexity can be reduced to $2NHWC^2/k^2 + CHW$ where $k$ is kernel size of the window for pooling. Further, we can use group-wise whitening to improve efficiency, reducing the cost of computing $\Sigma^{-\frac{1}{2}}$ from $TC^3$ to $TCg^2$ ($g$ is group size). For AII branch, we focus on the additional parameters introduced by two FC layers. In fact, the reduction ratio $r$ can be appropriately chosen to balance model complexity and representational power. Besides, the majority of these parameters come from the final block of the network. For example, a single AII in the final block of ResNet-50 has $2 * 2048^2/r$ parameters. In practice, the CE blocks in the final stages of networks are removed to reduce additional parameters. We provide the measurement of computational burden and Flops in Table 1.

**ResNet Training Setting**. All networks are trained using 8 GPUs with a mini-batch of 32 per GPU. We train all the architectures from scratch for 100 epochs using stochastic gradient descent (SGD) with momentum 0.9 and weight decay 1e-4. The base learning rate is set to 0.1 and is multiplied by 0.1 after $30, 60$ and 90 epochs. Besides, the covariance matrix in BD branch is calculated within each GPU. Since the computation of covariance matrix involves heavy computation when the size of feature map is large, a $2 \times 2$ maximum pooling is adopted to down-sample the feature map after the first batch normalization layer. Like (Huang et al., 2019), we also use group-wise decorrelation with group size 16 across the network to improve the efficiency in the BD branch. By default, the reduction ratio $r$ in AII branch is set to $4$.

**MobileNet V2 training Setting**. All networks are trained using 8 GPUs with a mini-batch of 32 per GPU for 150 epochs with cosine learning rate. The base learning rate is set to 0.05 and the weight decay is 4e-5.

**ShuffleNet V2 training Setting**. All networks are trained using 8 GPUs with a mini-batch of 128 per GPU for 240 epochs with poly learning rate. The base learning rate is set to 0.5 and weight decay is 4e-5. We also adopt warmup and label smoothing tricks.

**VGG networks on CIFAR10 training setting**. For CIFAR10, we train VGG networks with a batch size of 256 on a single GPU for 160 epochs. The initial learning rate is 0.1 and is decreased by 10 times every 60 epochs. The inhibited channel ratios in Fig. 1 and Fig.3(c) is measured by the average ratio for the first 6-th layers since the bottom layers can extract rich feature representation and the sparsity is not desired. For inference drop experiments in Fig.1(c), we randomly drop channels in the third layer with different dropout ratio. For each ratio, we run the experiment 5 times and average the top 1 accuracy.

**Mask-RCNN training setting in COCO**. We fine-tune the ImageNet pretrained model in COCO for 24 epoch with base learning rate 0.02 and multiply it by 0.1 after 16 and 22 epochs. All the models are trained using 8 GPUs with a mini-batch of 2 images. The basic backbone structure is adopted from the ResNet50/ResNet101 trained on ImageNet.

