# OpenReview forum: "Channel Equilibrium Networks"
_ICLR.cc/2020/Conference — Reject_

### Official Review · AnonReviewer3 · 2019-10-24
**Official Blind Review #3**

**Rating:** 3

**Review:**

The authors point out that CNNs can develop collapsed channels that limit their capacity. They propose to remedy this with batch decorrelation (BD), which focuses on ensuring that channels play an equal role in the feature map and are less likely to collapse. The claim is supported with experiments on CIFAR10, ImageNet and COCO.

Figures 1 and 3 show convincingly that BD reduces channel collapse. The overall results are modest improvements, but the point the authors make is that reducing collapse is correlated with improved performance.

I selected to reject the paper because in its current form a general readership will struggle to understand it. An expert familiar with the details of the field (not just the general area) can probably disentangle the text, but otherwise it's too convoluted.

An example note on math:
It would be great to reduce confusion and improve the consistency of math in the paper. In some parts of the manuscript, like section 3 "Notations", the symbols are well defined and clear to understand. In other locations, like section 1, an undefined \gamma is mentioned in the text, without definition. Is that \gamma the same as in section 3? What is its definition in section 1?

Assuming that \gamma is as (loosely) defined in section 3, the reader can go back to section 1. This is an example of the convolution I mention above. For someone who has read the paper (and already read section 3), the \gamma in section 1 makes sense.

**Experience Assessment:**

I do not know much about this area.

**Review Assessment: Checking Correctness Of Derivations And Theory:**

I assessed the sensibility of the derivations and theory.

**Review Assessment: Checking Correctness Of Experiments:**

I assessed the sensibility of the experiments.

**Review Assessment: Thoroughness In Paper Reading:**

I read the paper at least twice and used my best judgement in assessing the paper.

---

> ### Author Response · Authors · 2019-11-15
> **Response to AnonReviewer3**
>
> We thank the reviewer for the constructive feedback and thorough reading of our paper. We list detailed responses to some concerns.
> 1) “They propose to remedy this with batch decorrelation (BD).”
> In fact, we propose the CE block to remedy the inhibited channel, consisting of BD and AII branches. Fig.3(c) shows the inhibited channel ratio of plain BN, BD, and CE under different weight decays. As we can see, both BD and CE can reduce inhibited channel ratio significantly compared with plain BN. Table 4 further shows that BD and AII help with each other.
> 2) “The overall results are modest improvements.”
> Compared with other approaches in building block design [1,2], the proposed CE block improves performance by a large margin. For example, on ImageNet dataset, GCNet [1] with ResNet-50 as backbone improves the top-1 accuracy of baseline by 0.1, from 76.9 to 77.7 while CE-ResNet-50 improves it from 76.6 to 78.3. In COCO data-set, GCNet achieves 41.4 bounding box AP and 37.0 segmentation AP while CE achieves 42 bounding box AP and 37.5 segmentation AP, which is the best performance with ResNet-50 as the backbone to our best knowledge.
> 3) “But the point the authors make is that reducing collapse is correlated with improved performance.”
>  In the latest manuscript, we emphasize that CE achieves higher performance by encouraging all channels to contribute equally (channel equalization). Preventing inhibited channels is one of the attractive properties of CE. We have re-organized the introduction part to make our motivation more clear and intuitive (see the answer (1) in response to the AnonReviewer5 and introduction part).
> 4) “In its current form a general readership will struggle to understand it.”
> We are sorry about the unclear descriptions of our methods so we have made lots of modifications in the latest manuscript to make our paper more clear and neater. For example, we have put forward the formulation of ‘Norm+ReLU’ basic block in the introduction part. For clarity, we have also provided detailed explanations for some terminology such as channel equalization and equivalent $\gamma$.
> [1] GCNet: Non-local Networks Meet Squeeze-Excitation Networks and Beyond, CVPR2019
> [2] Grad-cam: Visual explanations from deep networks via gradient-based localization, ECCV, 2017

---

### Official Review · AnonReviewer1 · 2019-10-24
**Official Blind Review #1**

**Rating:** 6

**Review:**

This paper proposes a new block that gives diminishing

Pros)
(+) This paper is well-written and the idea looks interesting.
(+) The provided experiments look quite extensive. I like the experiments with both heavy and light network architectures to show the effectiveness of the proposed method.
(+) The authors provided plausible and sufficient backups for the claim.
(+) Connection with Nash equilibrium seems to be slightly overclaimed, but the attempt is novel and interesting.


Cons)
(-) The overall paper flow could be organized much better. I think some of the materials in the appendix needs to be placed in the main paper.
(-) Some of the notations in Section 3 need to be rewritten for better clarity.
(-) The terminology such as channel-level sparsity, channel equalization, and equivalent lambda looks vague. I recommend the authors define them for clarity.

Comments)
- I think the relationship between diminishing the sparsity on channels and improving the evaluation performance should be more addressed. If we change ReLU after Conv with other nonlinear functions such as ELU or PReLU, then the output hardly becomes zero, but as you know the performance usually gets worse. Then, how do you explain the necessity of reducing sparsities?
- How did you measure the sparsity in a plain network in Figure 1 and Figure 3? Did you pick a single layer or gather all the layers' sparsity? Please specify the way of measuring it.
- How did you plot the graphs in Figure 3.(a) and 3.(b). Did you pick a single channel randomly?
- To support the authors' claim, it would be better to show the sparsity ratio for a model (ResNet18, MobileNetV2, and so on) which shows better performance than each original one.
- How did you determine the transformation of F in the AII branch? Please provide any intuitions why the authors choose the transformation like in eq.(9).
- If CE-block actually conducts well right after a BN layer, then why CE block is attached only after the final BN layer in a bottleneck module? It would be better to provide any results by doing some studies when dealing with the intermediate BNs in a bottleneck module.
- I am wondering whether adaptive normalization techniques such as Instance-Batchnormalziation or Switchable Normalization methods could also give less sparsified features. Please clarify this by comparing with the proposed method.
- It seems that CE would have some extra computational costs compared with SE's. Please clarify why CE-module does have small computational costs.
- Please specify what MobileNetV2 has used. Looks like MobileNetV2x1.0 would be used.
- Why do you think the Top-5 accuracy of MobileNetV2 has not been improved?
- Can your method combine with ResNet with SE such as ResNet50SE?
- Please clarify the way of measuring the sparsity ratio in Figure 3d.   Why did you consider the sparseness of a feature by measuring the gamma in Figure 4? There may exist a channel that contains large magnitude values then gamma cannot make the output close to zero.

About overall rating)
The theoretical backups for the authors' claim look sound, and all the experiments including the performance improvement with several models seem to support the effectiveness of the idea very well. Specifically, I like all the analyses and the connection with Nash equilibrium, which are very intuitive and may provide further insights to researchers in this field.  I think this paper is clearly above the standard of ICLR. If the authors could address all the concerns above and refine the paper with better readability, then I could increase the score.

**Experience Assessment:**

I have published in this field for several years.

**Review Assessment: Checking Correctness Of Derivations And Theory:**

I assessed the sensibility of the derivations and theory.

**Review Assessment: Checking Correctness Of Experiments:**

I carefully checked the experiments.

**Review Assessment: Thoroughness In Paper Reading:**

I read the paper at least twice and used my best judgement in assessing the paper.

---

> ### Author Response · Authors · 2019-11-15
> **Response to  AnonReviewer1**
>
> We thank the reviewer for constructive feedback and helpful suggestions. We have provided detailed explanations for terminology and performed additional experiments to work towards addressing the concerns the reviewer has raised. The detailed responses to some concerns are listed below,
> 1) diminishing sparsity and improving performance
> Thanks for the suggestion. In revision, we emphasize that CE achieves higher performance by encouraging all channels to contribute equally to the network’s prediction (channel equalization). Preventing inhibited channels is one of the attractive properties of CE. The relationship between channel equalization and improving performance is established by extensive experiments in various ‘Norm+ReLU’ blocks (Fig.1(c) and Fig.6) and modern CNN architectures (Fig.3(a,b)).
> 2) sparsity in Fig. 1 and Fig. 3
> In revision, we refer to those less important channels as inhibited channels. In Fig.1 and Fig.3. the inhibited channel ratio is defined as the average percentage of values less than 1e-2 in the first six feature maps of VGGNet. We initially put these details in footnotes. We have now moved all footnotes containing key details into the main text.
> 3) How to plot Fig. 3(a&b)?
>  For all cumulative ablation curves, we randomly drop channels in the first normalization layer with different dropout ratio for all networks. For each drop ratio, we run the experiment 5 times and average top-1 accuracy.
> 4) “a model with better performance and lower sparsity ratio”
> In revision, our main motivation is that a network that generalizes well would have its channels contributed equally to the network’s prediction. We claim that channel equalization is important for a CNN to obtain good performance, which is demonstrated by a cumulative ablation method in terms of various architectures (VGGNet, ResNet-50, and MobileNet V2) and many ‘Norm+ReLU’ blocks (‘BN+ReLU’, ‘BN+ELU’ and so on).
> 5) “transformation of F in the AII branch?”
> We describe the motivation of designing F in revision. One of the requirements is that dependencies among channels should be embedded in F. Following the best practice in SENet, we construct F by a bottleneck architecture that is able to model channel dependencies and limit model complexity.
> 6) “Dealing with the intermediate BNs in a bottleneck module.”
> Thanks for your suggestion. Since the CE block is expected to help channel equalization, it would benefit from a larger number of channels. Therefore, we insert CE in a layer that consists of the largest number of channels in the bottleneck for all networks. For example, the top-1 acc decreases when CE is inserted in the ‘3x3’ convolution layer of a bottleneck for ResNet-50 (see detail in  Sec.E.2). Besides, if we plug CE into each ‘Norm+ReLU’ block of a network, it would involve heavy computation.
> 7) “ adaptive normalizers and sparsified features.”
> Thanks for your comment. In the latest manuscript, we emphasize that CE achieves higher performance by encouraging all channels to contribute equally (channel equalization). Hence, the channel equalization of BN, SN, and CE are compared in Fig.6(b). SN basically do damage to channel equalization compared with BN. However, CE can improve it a lot.
> 8) “computational costs compared with SE's”
> CE consists of two branches, i.e. BD and AII. CE has comparable computational cost in inference for two reasons. (1) BD can be merged into convolutional layers in inference, which adds extra computation marginally. (2) CE is not used in convolution layers with too many channels. For example, CE is not employed in the last bottleneck for ResNet-50 while SE is inserted into each bottleneck. Table 1 shows that CE-ResNet50 only increases GFLOPs by 0.24% compared with SE-ResNet50.
> 9) “type of MobileNet V2”
>  MobileNet V2 1.0x is used. We have specified it in Table 2.
> 10) ”the Top-5 and top-1 acc”
> We have carefully checked the trained model and confirm that this is indeed the truth. As we can see, compared with SE, CE achieves 1.1 higher top-1 accuracy while the same top-5 accuracy. We guess that CE is better at accurately predicting the target label for images.
> 11) “ combine with SE-ResNet”
> Thanks for your suggestion. CE encourages all channels to contribute equally to feed-forward representation and SENet selectively emphasizes informative channels and suppresses less useful ones.  If we first equalize channels via CE block and then employ SE to learn information specific to each channel, it would make channels both equalized and distinctive. This could be an extension for future study.
> 12) “sparsity ratio in Fig.3(d)” and “sparseness measured by gamma in Fig. 4”
> We are sorry for the wrong description of the sparsity ratio in Fig.3(d) on page 8 of the previous version. Please see the answer to Question 2 for more details. In addition, Fig.4 shows the principal components of the input of AII sub-network get a lower variance, suggesting the input feature has a more valid basis and thus more informative.

---

### Official Review · AnonReviewer5 · 2019-10-31
**Official Blind Review #5**

**Rating:** 3

**Review:**

This paper studies the channel-collapsed problem in CNNs using 'BN+ReLU' . The Channel Equilibrium block which consists of batch decorrelation branch and adaptive instance inverse branch are proposed to reduce the channel-level sparsity. Experiments on ImageNet and COCO demonstrate that the proposed CE block can achieve higher performance than the conventional CNNs by introducing little computational complexity. The author also discuss the relationship between the proposed method and Nash Equilibrium.

Pros:

+ The experimental results are impressive, the proposed block can improve the accuracy of CNNs while requires little additional computation cost.
+ This paper is well-written and easy to follow. The authors give a explicit explanation as well as prove of the proposed scheme.

Cons:

- The motivation of this paper seems to be weak. The author argues that popular CNNs with 'BN+ReLU' have certain channels which would always output 0 for any input. Why not directly remove this channel to achieve speed-up?
- Moreover, the author argues that 'BN+ReLU' block would lead to channel-level sparsity according to [1]. However, [1] says that this sparsity relies on weight decay. Figure 3 (d) also proves that the sparsity ratios of BN and CE are all 0 when weight decay is set as 0 (also notes that they achieve best accuracy when weight decay is 0). The results demonstrate that the higher accuracy of CE does not rely on its lower sparsity ratio.

In conclusion, the proposed CE is effective for achieving higher accuracy. However, the motivation and argument of the proposed method seems to be invalid, which prevents this work to be accepted.

[1] On implicit filter level sparsity in convolutional neural networks. CVPR, 2019.

**Experience Assessment:**

I have published in this field for several years.

**Review Assessment: Checking Correctness Of Derivations And Theory:**

I carefully checked the derivations and theory.

**Review Assessment: Checking Correctness Of Experiments:**

I carefully checked the experiments.

**Review Assessment: Thoroughness In Paper Reading:**

I read the paper at least twice and used my best judgement in assessing the paper.

---

> ### Author Response · Authors · 2019-11-15
> **Repsonse to AnonReviewer5**
>
> We thank AnonReviewer5 for the kind comment and helpful feedback. We also appreciate that the reviewers commented that “the motivation seems to be weak” and we are working on it accordingly. Here is the detailed response to the technical comments.
> 1) “The motivation of this paper seems to be weak.”
> We are sorry for the vague narration of the motivation. We have made lots of modifications in the latest manuscript to make the motivation more clear and neater. Specifically, a  key motivation is that a network with good generalization ability usually has its channels contributed equally to its feed-forward computation [1]. However, We observe that the inhibited channels (smaller values of features than other channels) exist in many “norm+ReLU-like” basic blocks such as ‘BN+ReLU’, ‘LN+ReLU’, and ’BN+ELU’[2], as shown in Fig.1(a,b).  The inhibited channels are inactive during training and cause disequilibrium among channels,  leading to limited learning capacity and generalization ability. Therefore, we propose the Channel Equilibrium (CE) block to equalize the channels’ magnitudes by applying a decorrelation operator. This operator enables all the channels to play an equal role in the computations of a CNN, improving the generalization ability of the network.  In this way, the introduction part has been re-organized. Please see the detailed changes in Sec.1.
> 2) “Why not directly remove these channels to achieve speed-up?”
> A lot of work has investigated in removing the inhibited channels that are less important to the network’s prediction, such as work in network slimming.  However, simply removing the inhibited channels that are inactive during training does not help improve learning capacity.  This work presents an alternative perspective that encourages all channels to contribute equally to the feature representation learning process by proposing the CE block. CE improves the learning capacity and generalization ability of many networks,  achieving state-of-the-art results on various challenging benchmarks such as ImageNet and COCO.
> 3) “Figure 3 (d) also proves that the sparsity ratios of BN and CE are all 0 when weight decay is set as 0. The results demonstrate that the higher accuracy of CE does not rely on its lower sparsity ratio.”
> We did not mark the coordinates of all points due to the limited space. The weight decay of the first point is not $0$ but $1e-3$. We have carefully checked the trained model and found that the sparsity ratio is very small but not $0$.  In revision, we refer those less important channels as ‘inhibited channels’ and inhibited channel ratio is defined as the average percentage of values less than 1e-2 in the feature maps of the network. Fig.3(c,d) are re-plotted. The inhibited channel ratio of BN, BD, and CE are 11.26%, 3.02%, 1.52%,  respectively, showing that CE achieves higher accuracy and prevents inhibited channels. Note that we clarify that preventing the inhibited ratio is one of the properties of CE in the revised version.
> 4) “the motivation and argument of the proposed method seem to be invalid.”
> In the latest manuscript, we emphasize that CE achieves higher performance by encouraging all channels to contribute equally (channel equalization), which is verified by extensive experiments like Fig.1(c), Fig.3(a&b), and Fig.6. One of the attractive properties of CE is preventing inhibited channels, which is demonstrated in both experiments (Fig.1(a&b) and  Fig.3(c)) and theory (Proposition 1).
> [1] On the importance of single directions for generalization. ICLR, 2018.
> [2]  On implicit filter level sparsity in convolutional neural networks. CVPR, 2019.

---

### Author Response · Authors · 2019-11-15
**General Response to reviewers accompanying revision**

We are thankful to the reviewers for their careful check and thoughtful suggestions on our previous manuscript. In particular, we are glad that the reviewers agreed that “the experimental results are impressive”, that “the theoretical backups for the authors’ claim look sound” and that “the attempt that connects with Nash Equilibrium is very intuitive and may provide further insights”.

To address the reviewers’ concerns, we tried our best to clarify our motivation, improve language usage, give more analysis and explanations in preparing the revised version. Specifically, we have performed several additional experiments, including additional figures detailing our motivation with more clear narration and demonstrating the superiority of the proposed CE block. For detailed descriptions of the changes we have made, please find the itemized responses to individual reviewers below. As a result of these changes, our paper is now more than nine pages long. Due in large part to the reviewers’ constructive feedback, we believe that our paper has been substantially strengthened.

---

### Decision · Program_Chairs · 2019-12-19

**Decision:**

Reject

**Comment:**

The paper proposed Channel Equilibrium (CE) to overcome the over-sparsity problem in CNNs using 'BN+ReLU'. Experiments on ImageNet and COCO show its effectiveness by introducing little computational complexity. However the reviewers pointed a number of problems in the writing and the clarity of the paper. Although the authors addressed all the se concerns in details and agreed to make revisions in the paper, it's better for the authors to submit the revised version to another opportunity.